# Comparison of Virulence-Factor-Encoding Genes and Genotype Distribution amongst Clinical *Pseudomonas aeruginosa* Strains

**DOI:** 10.3390/ijms24021269

**Published:** 2023-01-09

**Authors:** Tomasz Bogiel, Dagmara Depka, Stanisław Kruszewski, Adrianna Rutkowska, Piotr Kanarek, Mateusz Rzepka, Jorge H. Leitão, Aleksander Deptuła, Eugenia Gospodarek-Komkowska

**Affiliations:** 1Microbiology Department, Ludwik Rydygier Collegium Medicum in Bydgoszcz, Nicolaus Copernicus University in Toruń, 85-094 Bydgoszcz, Poland; 2Clinical Microbiology Laboratory, Doctor Antoni Jurasz University Hospital No. 1 in Bydgoszcz, 85-094 Bydgoszcz, Poland; 3Institute of Forensic Genetics Ltd., 85-071 Bydgoszcz, Poland; 4Medicover Integrated Clinical Services, 85-065 Bydgoszcz, Poland; 5Department of Microbiology and Food Technology, Faculty of Agriculture and Biotechnology, Bydgoszcz University of Science and Technology, 85-029 Bydgoszcz, Poland; 6Department of Bioengineering, IBB—Institute for Bioengineering and Biosciences, Instituto Superior Técnico, Universidade de Lisboa, Av. Rovisco Pais, 1049-001 Lisboa, Portugal; 7Associate Laboratory i4HB—Institute for Health and Bioeconomy at Instituto Superior Técnico, Universidade de Lisboa, Av. Rovisco Pais, 1049-001 Lisboa, Portugal; 8Department of Propaedeutics of Medicine and Infection Prevention, Ludwik Rydygier Collegium Medicum in Bydgoszcz, Nicolaus Copernicus University in Toruń, 85-094 Bydgoszcz, Poland

**Keywords:** multiple-antibiotic resistance, *Pseudomonas aeruginosa* virulence genes, virulence-factor-encoding genes, virulence gene genotyping, *Pseudomonas aeruginosa* clinical isolates, antibiotic resistance profiles

## Abstract

*Pseudomonas aeruginosa* is an opportunistic pathogen encoding several virulence factors in its genome, which is well-known for its ability to cause severe and life-threatening infections, particularly among cystic fibrosis patients. The organism is also a major cause of nosocomial infections, mainly affecting patients with immune deficiencies and burn wounds, ventilator-assisted patients, and patients affected by other malignancies. The extensively reported emergence of multidrug-resistant (MDR) *P. aeruginosa* strains poses additional challenges to the management of infections. The aim of this study was to compare the incidence rates of selected virulence-factor-encoding genes and the genotype distribution amongst clinical multidrug-sensitive (MDS) and MDR *P. aeruginosa* strains. The study involved 74 MDS and 57 MDR *P. aeruginosa* strains and the following virulence-factor-encoding genes: *lasB*, *plC H*, *plC N*, *exoU*, *nan1*, *pilA,* and *pilB*. The genotype distribution, with respect to the antimicrobial susceptibility profiles of the strains, was also analyzed. The *lasB* and *plC N* genes were present amongst several *P. aeruginosa* strains, including all the MDR *P. aeruginosa*, suggesting that their presence might be used as a marker for diagnostic purposes. A wide variety of genotype distributions were observed among the investigated isolates, with the MDS and MDR strains exhibiting, respectively, 18 and 9 distinct profiles. A higher prevalence of genes determining the virulence factors in the MDR strains was observed in this study, but more research is needed on the prevalence and expression levels of these genes in additional MDR strains.

## 1. Introduction

*Pseudomonas aeruginosa* is a ubiquitous Gram-negative γ-proteobacterium commonly isolated from soil, water, plants, feces, compost, fungi, and sediments. It is also a commensal bacterium transiently inhabiting the digestive tract and skin of humans and animals. Due to its high genomic plasticity and metabolic versatility, *P. aeruginosa* is able to colonize and cause infection in a wide range of hosts, from unicellular organisms to humans [1,2,3,4].

*P. aeruginosa* is responsible for hospital-associated infections (HAIs) as an opportunistic pathogen, posing a serious risk to patients affected by a wide range of diseases, such as cystic fibrosis, immune deficiencies, cancer, chronic pneumonia, ventilator-associated pneumonia (e.g., during the COVID-19 pandemic), and burn and surgical wounds infections. *P. aeruginosa* is one of the most common etiological agents of intensive-care-unit (ICU)-acquired pulmonary, urinary, and bloodstream infections [3,5,6]. The pathogenesis of the infections caused by *P. aeruginosa* is associated with multiple factors leading to effective colonization and biofilm formation, tissue necrosis, invasion, and dissemination through the vascular system, as well as the activation of both local and systemic inflammatory responses [7].

The presence of diverse properties encoded in its genome, namely virulence and antimicrobial-resistant determinants, led to the classification of *P. aeruginosa* as a member of the ESKAPE group (amongst the *Enterococcus faecium*, *Staphylococcus aureus*, *Klebsiella pneumoniae*, *Acinetobacter baumannii*, and *Enterobacter* species) [4,8]. The occurrence of multidrug-resistant (MDR) *P. aeruginosa* strains is a significant medical issue. MDR *P. aeruginosa* isolates, in addition to their intrinsic intracellular resistance mechanisms (such as the overexpression of the efflux pumps and low outer membrane permeability), are capable of exogenously acquiring genes related to the synthesis of enzymes inactivating β-lactams, as well as genes encoding antimicrobial efflux system pumps [9,10].

In *P. aeruginosa*, the genes encoding virulence factors may be located on the bacterial chromosome or on the additional genome–genetic elements, such as the plasmids and pathogenicity islands. Virulence factors can be categorized as extracellularly secreted and bacterial-cell-associated. The pathogenicity of *P. aeruginosa* depends on the interaction of multiple virulence factors in each stage of an infection [11]. The synthesis of enzymes (such as elastases, phospholipases, and neuraminidases), extracellular toxins (e.g., exotoxins T, U, and Y), and cell components (e.g., pili) are some of the virulence factors that can be identified [12].

A detailed understanding of the contributions of the genes that encode virulence factors in both MDR and multidrug-sensitive (MDS) strains allows for a better understanding of the evolutionary strategy and pathogenicity of *P. aeruginosa*. This knowledge could also contribute to the effective control of the pathogen’s spread, especially in hospital environments, where the presence of virulent MDR strains is a critical threat to patients’ health and lives [13].

The aim of this study was to compare the incidence of virulence-factor-encoding genes and the genotype distribution amongst clinical MDR and MDS *P. aeruginosa* strains. The study involved 74 MDS and 57 MDR *P. aeruginosa* strains of clinical origin, and the presence of the virulence-factor-encoding genes *lasB*, *plC H*, *plC N*, *exoU, nan1*, *pilA,* and *pilB* in their genomes was investigated. Their consecutive genotype distribution, with respect to the antimicrobial susceptibility profiles of the strains, was also analyzed. The mentioned genes were selected based on previous studies in order to analyze their presence among clinical strains with different phenotypic susceptibility profiles.

## 2. Results

### 2.1. The Origin and Prevalence of Virulence-Factor-Encoding Genes among MDR and MDS P. aeruginosa Strains

In total, 74 MDS and 57 MDR clinical isolates of *P. aeruginosa* strains were included in the study. Their origins are presented in Figure 1 and Figure 2 (MDS) and Figure 3 and Figure 4 (MDR).

The assessment of the virulence genes frequency revealed a wide variety in the genes’ distribution. The incidence of virulence genes amongst the examined strains was as follows: the *lasB* and *plC N* genes were detected in all the tested MDR strains, whereas the *pilA* and *pilB* genes presented with the lowest frequency amongst both groups of strains, being around 5.0% and below 2.0% for *pilA* and *pilB*, respectively. The incidence of the detected virulence genes amongst the examined *P. aeruginosa* strains is shown in Figure 5, while their detailed incidence is provided in the Appendix A (Appendix A for the MDS strains and Appendix A for the MDR isolates).

There was a statistically significant relationship between the frequency of the *lasB* (*p* = 0.0354), *plC N* (*p* = 0.0051) and *plC H* (*p* = 0.0227) genes and their occurrence among the MDR isolates.

In the group of MDR strains, the only moderate positive correlation observed was that of the *pilA* and *pilB* genes (r = 0.5669). In the group of MDS strains, the only moderate positive correlation observed was that of the *plC H* and *lasB* genes (r = 0.6752), and a weak positive relationship between the *plC H* and *plC N* gene was noted (r = 0.2849). In the present study, the only negative correlation between the genes, in terms of their presence, was noted among the MDS strains (r = −0.2327, weak) in the case of the *exoU* and *pilA* genes. The detailed statistical calculation of Spearman’s rank correlation coefficient for the particular gene pairs, with respect to the antimicrobial susceptibility profiles of the *P. aeruginosa* strains included in the study, is presented in Appendix A.

### 2.2. Prevalence of the Observed Genotypes

Eighteen genotypes were observed (named I-S to XVIII-S) in the group of MDS *P. aeruginosa* strains. Their prevalence and distribution amongst the examined strains are shown in Table 1. The most prevalent genotype, numbered VII-S, including the *lasB*, *plC H*, and *plC N* genes, was observed among 22 (29.7%) of the isolates. Eight (10.8%) strains presented individual genotypes. The incidence of the genotypes identified amongst the studied *P. aeruginosa* MDS strains is presented in Table 1, while the detailed specification of their genotypes is presented in the Appendix A.

Nine genotypes (named I-R to IX-R) were observed among the MDR *P. aeruginosa* strains. Their prevalence and distribution amongst the examined strains are shown in Table 2. The most prevalent genotype, named III-R, consists of all the genes detected (except for the *nan1*, *pilA*, and *pilB* genes) and was observed among 20 (35.1%) of the isolates, while the VII-R genotype, including the *lasB*, *plC H*, and *plC N* genes, was noted in 19 (33.3%) of the strains. Five (8.8%) strains presented individual genotypes. The incidence of the genotypes amongst the *P. aeruginosa* MDR strains is presented in Table 2, while the detailed specification of the genotypes is shown in the Appendix A. However, no statistically significant correlation was found between the presence of particular genotypes in the MDS or MDR groups of strains (χ^2^ = 1.2375, *p* = 0.7440).

## 3. Discussion

For many years, *P. aeruginosa* strains have frequently been the object of many studies, mostly due to their relevance for hospital-acquired infections. The high incidence of MDR *P. aeruginosa* in HAIs and infections among ICU patients renders the constant observation of this pathogen necessary [14]. The global threat is significant, since the combination of antibiotic resistance mechanisms and virulence factors usually determines the severe course of the infection [9].

To the best of our knowledge, the present study was one of the largest in terms of the number of the strains included in the study and the number of diverse virulence factors investigated. The detected genes were selected to cover a number of virulence factors with different activities, e.g., typical toxins, the most important enzymes, and structure-dependent factors, including well-known and rarely investigated genes.

Elastase B, encoded by the *lasB* gene, is an important virulence factor with a widespread distribution among *P. aeruginosa* strains. Through the hydrolysis of antibodies and complement components, this metallopeptidase influences the disintegration of intercellular connections in the host tissues and interferes with the process of normal immune responses [15]. In this study, the elastase B-encoding gene was present in all the MDR strains analyzed. The gene was also present in high proportions among the MDS strains (91.9%). A study by Elmouaden et al. [16] also reported high proportions of the *lasB* gene among both MDR strains (99%) and MDS strains (97.6%). In addition, Ratajczak et al. [17] reported the leading contribution of the *lasB* gene among the majority of the *P. aeruginosa* strains tested (93.1%). Similar findings were reported in the study conducted by Dawodeyah et al. [18], which revealed that all the MDR strains contained the *lasB* gene. The results obtained in our study are in line with the general pattern of the *lasB* gene’s ubiquitous distribution in *P. aeruginosa* and its identification as a significant predictor of the species’ evolutionary strategy. It is also worth noting the high fraction of this gene among non-clinical strains and the spread of its presence in both natural and anthropogenic contexts [19,20].

In our study, a moderately positive correlation was found between the presence of the *lasB* gene and the *plC H* genes in MDS strains. Both the hemolytic (*plC H*) and the non-hemolytic phospholipase C (*plC N*) play an important role in the pathogenesis of *P. aeruginosa* infections. The products of these bacterial enzymes compete with the second messengers of the host enzymes, affecting physiological cell processes and disrupting cell membrane lipid homeostasis [21]. There is a noticeable difference in the number of publications on phospholipase genes, with the hemolytic phospholipase described much more frequently. This study identified the *plC N* gene in 100% of the MDR strains and in 87.8% of the MDS strains, while the gene encoding the synthesis of hemolytic phospholipase C was identified in 96.5% and 83.8% of the strains, respectively. The results are quite similar to the values obtained in our previous study on bloodstream-derived isolates, where the presence of the *plC N* and *plC H* genes was detected in 95.8% and 100% of the strains, respectively [22]. Meanwhile, Gonçalves et al. [23] identified a lower prevalence of the *plC N* gene among carbapenem-resistant strains (88.0%) compared to the results of this work. Additionally, the study by Wolska and Szweda [24] identified a higher prevalence of the *plC H* gene (95.2%) compared to the *plC N* gene (88.7%). Despite varying reports regarding the proportions of individual genes encoding phospholipase C synthesis, it should be concluded that these genes encode one of the most common virulence factors, regardless of the susceptibility patterns of the *P. aeruginosa* strains. The frequent detection of both virulence genes among the strains demonstrates that these genes may serve as potent factors in the development of infections and that their presence affects the population dynamics of *P. aeruginosa*. Notably, this study also found a weak positive correlation between the presence of both discussed phospholipases genes.

The presence of exotoxin U (encoded by the *exoU* gene), as the most toxic of the four effector proteins observed in the pathogenesis of *P. aeruginosa* infections, results in a more severe disease course. This effector protein induces a highly rapid lytic effect on the host cell membrane, leading to necrosis in the epithelial cells, macrophages, and neutrophils [25]. This study identified a moderate level of presence of the *exoU* gene in both the MDR and MDS strains, which occurred in 52.6% and 48.6% of the strains, respectively. A lower incidence was reported by Naik et al. [26], where the distribution of the *exoU* gene among MDR strains was 8.7%, while the aforementioned gene was absent in MDS. It is worth noting that the study also referred to the presence of the *exoS* and *exoA* genes, and the proportions of the genes among the genotypes varied significantly. The investigation by Mitov et al. [27] identified the presence of the *exoU* gene in 30.2% of the 202 strains tested, while the percentage of this gene among the strains isolated from cystic fibrosis patients was 28.6%, slightly below the level of 30.6% for the isolates from other origins. Horn et al. [28] studied a total of 189 clinical isolates and concluded that 22.8% of the strains carried the *exoU* gene, which was additionally associated with the multidrug resistance profiles of the strains. Moreover, Subedi et al. [29] also observed higher antibiotic resistance in *exoU*-positive strains. This stands in opposition to the results of our study, where the prevalence of this gene was similar in both the MDR and MDS strains.

Neuraminidase is also an important virulence factor involved in the pathogenesis of infections caused by *P. aeruginosa* [30]. The enzyme plays an integral role in respiratory tract infections by modifying receptors on the surfaces of the host cells and enabling mucosal colonization. Neuraminidase also affects cystic fibrosis pathogenesis, as it is involved in the bacterial colonization of mucins in the airways. The exact role of neuraminidase in the pathogenesis of infections is unexplored; however, *nan1*-negative *P. aeruginosa* strains fail to cause infections in in vivo models [31]. Our study reported the presence of the *nan1* gene in 28.1% of the MDR and in 31.1% of the MDS strains. Our previous study identified the prevalence of the *nan1* gene at a much higher level, occurring among 47.9% of the strains isolated from bloodstream infections, with variable levels of resistance to selected antibiotics [22]. It is noteworthy that the clinical samples in the previous study were also derived from patients at the same hospital. The absence of significant differences between the studies suggests a moderate dynamic of gene distribution in the local *P. aeruginosa* population [22]. Meanwhile, the study by Elmaraghy et al. [32] reported a higher presence of the *nan1* gene among MDR (46.4%) compared to MDS strains (31.6%), which stands in opposition to this study. The *nan1* gene was one of the least prevalent virulence genes among the studied strains in the cited study. As shown previously, the occurrence of the *nan1* gene is strongly influenced by the type of material from which the strains are isolated. The study by Nikbin et al. [33] identified the *nan1* gene in 4% of the *P. aeruginosa* strains derived from burn patients, in 30% of the strains isolated from infected wounds, and in 46.6% of the strains isolated from patients with respiratory tract infections. The second value was very close to the one obtained in our study, regardless of the susceptibility profiles of the investigated strains. The study by Mitov et al. [27] also noted the differential occurrence of the *nan1* gene, depending on the origin of the clinical material. These authors found that the *nan1* gene was most prevalent among the strains isolated from blood samples (62.5%), which was not an aspect of our study due to the high diversity of the strains’ origins. In contrast to the results of our study, these authors also demonstrated a statistically significant difference in the occurrence of the *nan1* gene between the MDR and MDS strains, being 40.2% and 13.2%, respectively [27]. This observation additionally illustrates differential trends in the gene’s distribution [27].

The distributions of the *pilA* gene among the MDR and MDS *P. aeruginosa* strains were similar, while the least frequent gene in the entire study population was the *pilB* gene. In addition, a positive, moderate correlation was found in regard to the occurrence of both genes in the MDR isolates. The tendencies of our results are opposite to those of the study conducted by Haghi et al. [34], where the level of the *pilA* gene’s prevalence among the MDR strains was much higher (24.7%) when compared to the *pilB* gene’s presence (17.2%). In addition, the study of the gene distribution among strains isolated from meat samples similarly confirmed the lower distribution of the *pilA* gene compared to *pilB*, corresponding to 6.9% and 24.1%, respectively [35]. This may be supported by the fact that the gene has a relatively low distribution in the *P. aeruginosa* population, regardless of the clinical or environmental origin. However, more studies on the differences between strains and their sites of isolation are required. Moreover, in the present study, the antimicrobial-sensitive strains demonstrated a weak negative correlation, *r* = −0.232670, between *pilA* and *exoU* genes in terms of their presence.

According to our research, the MDR strains harbored more virulence genes than the MDS strains. These results stand in contrast with those reported by Deptuła et al. [36], who found that the antimicrobial-resistant strains were less virulent than their drug-sensitive counterparts. However, as the authors noted, the study should be confirmed by the implementation of molecular biology techniques. Our previous studies on the determination of the presence of the exoenzyme S-encoding gene in *P. aeruginosa* strains with different antibiotic sensitivities also indicated the predominance of this gene in the MDS strains (without a statistically significant difference) [37]. On the other hand, in their study, Sonbol et al. [38] showed a correlation between the occurrence of virulence genes and the resistance mechanisms. Other reports also suggest the high plasticity of the *P. aeruginosa* genome, providing evidence for the replacement of genes encoding virulence factors with those showing antibiotic resistance or the opposite [16,17,18].

The diverse distribution of virulence profiles among the tested strains allowed for an analysis of the dominant genotypes. The dominant genotype among the MDR strains was the III-R genotype, covering 35.1% of the isolates. The III-R genotype contained the *lasB*, *plC N*, *plC H*, and *exoU* genes. Among the MDS strains, 29.7% of the strains belonged to the genotype VII-S, which contained the *lasB*, *plC N*, and *plC H* genes. This result confirms the previously discussed versatility of the occurrence of these genes, regardless of the origins and resistance profiles of the investigated isolates. The study by Hassuna et al. [39] also indicated high proportions of *lasB* and *exoU*, additionally identifying the simultaneous occurrence of the *aprA* and *exoS* genes in different configurations of the genomes of the studied MDR clinical *P. aeruginosa* strains. Meanwhile, the results obtained by Gajdács et al. [40] showed no correlation between the presence of the virulence genes and antibiotic resistance profiles when the studied virulence factors were the pigments and motility of the strains. On the other hand, a study by Hwang et al. [41] demonstrated the non-virulence of the MDR strains of *P. aeruginosa*. The diverse scientific reports on this subject may result from variations in the origins of the strains (clinical vs. environmental), the types of clinical material, or the geographic localization. The continuous monitoring of the dynamics of gene ratio changes in the *P. aeruginosa* strains may contribute to the development of an effective predictive tool based on the most common genes in both the MDR and MDS strains [34,39,41].

## 4. Materials and Methods

### 4.1. The Examined Strains

In total, 74 MDS and 57 MDR clinical isolates of *P. aeruginosa* strains isolated between 2015 and 2019 were included in the study. The clinical samples were treated using a standard procedure of microbiological diagnostic investigation in our unit. All the samples were initially plated on Columbia agar base supplemented with 5% sheep blood, MacConkey agar, Sabouraud agar (all from Becton Dickinson, Franklin Lakes, NJ, USA), and Cetrimide agar (bioMérieux, Marcy L’Etoile, France). The cultures were kept overnight at 37 °C.

An initial identification was conducted based on the bacterial morphology (Gram stain), performed directly using specimens derived from purulent material and respiratory tract and pre-culture blood samples with a BacTec FX device (Becton Dickinson).

The culture-based methods revealed a typical growth of the bacteria on the selective medium and the biochemical reaction results (oxidase and catalase). The final identification was performed using the MALDI-TOF MS method, performed using a MALDI Biotyper device (Bruker, Mannheim, Germany). All the strains were isolated and collected in the Clinical Microbiology Laboratory of Doctor Antoni Jurasz University Hospital No. 1 in Bydgoszcz, Poland. After the identification step, the strains were stored in brain heart infusion with 20% glycerol at −80 °C.

The strains were isolated from various clinical samples derived from different patients and additionally checked for their similarity using pulsed-field gel electrophoresis. Our goal was to exclude undistinguishable restriction patterns and avoid the investigation of repeated isolates. The *P. aeruginosa* strains ATCC 27853 (obtained from the American Type Culture Collection) and PAO1 isolate (kindly provided by the National Medicines Institute in Warsaw, Poland) were used as PCR positive controls for the investigation of particular genes.

### 4.2. Antimicrobial Susceptibility Testing and Strain Selection Criteria

During the antimicrobial susceptibility testing (AST) step, the following methods were used: the disc diffusion method on Mueller–Hinton agar (Becton Dickinson) for ticarcillin/clavulanate, with microdilution for the colistin MIC evaluation tests (SensiTest Colistin, Liofilchem, Roseto degli Abruzzi, Italy) and NMIC-402 panels of a Phoenix^TM^ M50 device (Becton Dickinson) for the remaining antimicrobials. Although the strains were isolated between 2015 and 2019, the results of the AST were interpreted according to the current European Committee on Antimicrobial Susceptibility Testing recommendations (EUCAST, Breakpoint tables for bacteria, Clinical breakpoints—bacteria v 12.0, 2022) in order to unify the strains’ susceptibility categories.

The strains that were phenotypically resistant to at least one of the representatives of at least three independent groups of antimicrobials were assigned to the MDR, as suggested by international experts of the European Centre for Disease Prevention and Control and the Centers for Disease Control and Prevention [42]. Those not fulfilling these criteria were assigned as MDS (sensitive to most of the antimicrobial groups).

### 4.3. Bacterial DNA Isolation

A Genomic Mini kit (A&A Biotechnology, Gdynia, Poland) was used for the DNA isolation, performed according to the manufacturer’s protocol. All the DNA samples were stored at −20 °C before their further use according to the purpose of the study.

### 4.4. Virulence Factors Genes Detection

The prevalence of 7 virulence-factor-encoding genes was determined by PCR in a separate reaction for each gene. The genes were amplified with primers selected on the basis of the published PAO1 strain genome sequence and the *P. aeruginosa* ATCC 27853 isolate, and the amplification procedure was carried out as previously described [43,44,45]. The reactions were performed in 0.2 mL test tubes (Eppendorf, Hamburg, Germany) with a final volume of 20 μL. Briefly, *Taq* polymerase was used with a total activity of 1 U per reaction in 1× concentrated polymerase buffer with 1.5 mM MgCl_2_ (Go *Taq* G2 Polymerase, Promega, Germany, FirePol DNA Polymerase, Solis BioDyne, Tartu, Estonia) and 200 μM deoxynucleotide triphosphates (Promega, Walldorf, Germany, Solis BioDyne, Tartu, Estonia). The primers were used at the final amount of 12.5 pmol per reaction (Integrated DNA Technologies, Coralville, IA, USA, Sigma, Darmstadt, Germany or Genomed, Warsaw, Poland). The primer sequences and the PCR annealing temperatures for each gene amplification are presented in Table 3. The DNA isolated from the *P. aeruginosa* PAO1 strain and the *P. aeruginosa* ATCC 27853 isolate served as the PCR positive controls. In the amplification procedure, a thermal cycler GeneAmp^®^ PCR System 2700 (Applied Biosystems, Foster City, CA, USA) was used. The presence of amplicons in the particular genes was evaluated visually through the application of gel electrophoresis (1.5% agarose, Bio-Rad, Feldkirchen, Germany) based on the product size and control strain DNA amplification (Appendix A). For the arbitrarily selected samples, PCR duplicates were performed to confirm the repeatability of the results, providing consistent results in each case.

### 4.5. Statistical Methods

Spearman’s rank correlation coefficient was calculated to investigate the correlations between particular genes, in terms of their presence, in a group of MDS and MDR strains (α = 0.05). Fisher’s two-rank exact test was applied to investigate the differences between the presence of particular genes in a group of MDS and MDR strains (α = 0.05). The chi-square (χ^2^) test was used to investigate the statistical significance of the correlation between the presence of a particular genotype in a group of MDS or MDR strains (α 0.05). Due to the high diversity of the genotypic compositions, this calculation was performed only for the genotypes represented by at least 11 strains in the whole group of the examined *P. aeruginosa* population.

All the calculations were performed using the STATISTICA 13.1 (data analysis software system, Poland) program of StatSoft, Inc., Cracow, Poland (2017).

## 5. Conclusions

This study found a higher prevalence of genes determining virulence factors in the MDR strains, but more research is needed on the prevalence and expression levels of these genes in MDR strains. The presence of *P. aeruginosa* can be confirmed using the broad dissemination of these genes in the study population. However, the presence of the *lasB* and *plC N* genes amongst the majority of the *P. aeruginosa* isolates, including all the MDR *P. aeruginosa*, might be used as a specific marker for diagnostic purposes. Overall, a higher diversity of genotypes was found among the MDS strains (18 profiles) when compared to the MDR strains (nine genotypes). The monitoring of the variability in the resistance and virulence genotypes can provide a useful tool with which to control *P. aeruginosa* infections more effectively, at least in the case of those caused by the strains presenting the MDR phenotype.

## Figures and Tables

**Figure 1 ijms-24-01269-f001:**
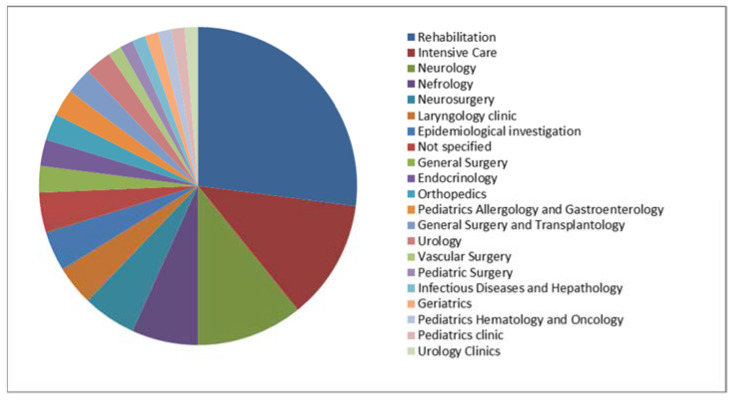
The origins (units) of the multidrug-sensitive *P. aeruginosa* strains included in the study (*n* = 74).

**Figure 2 ijms-24-01269-f002:**
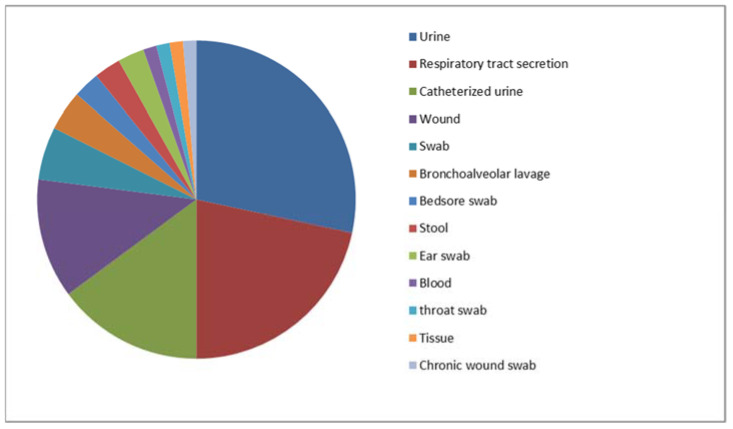
The origin (specimen types) of the multidrug-sensitive *P. aeruginosa* strains included in the study (*n* = 74).

**Figure 3 ijms-24-01269-f003:**
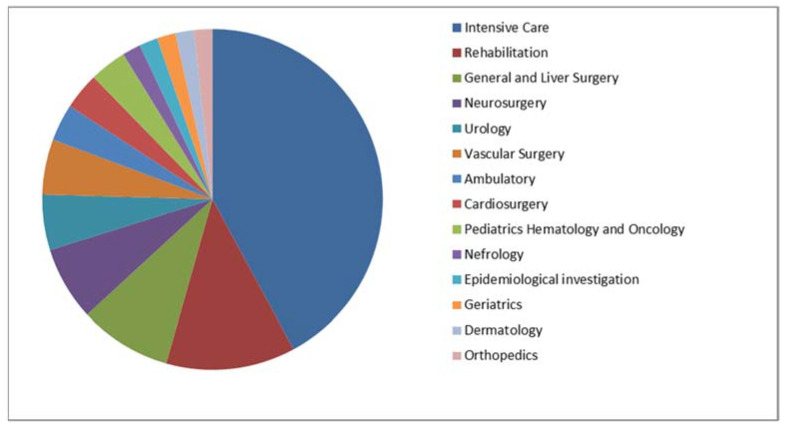
The origins (units) of the multidrug-resistant *P. aeruginosa* strains included in the study (*n* = 57).

**Figure 4 ijms-24-01269-f004:**
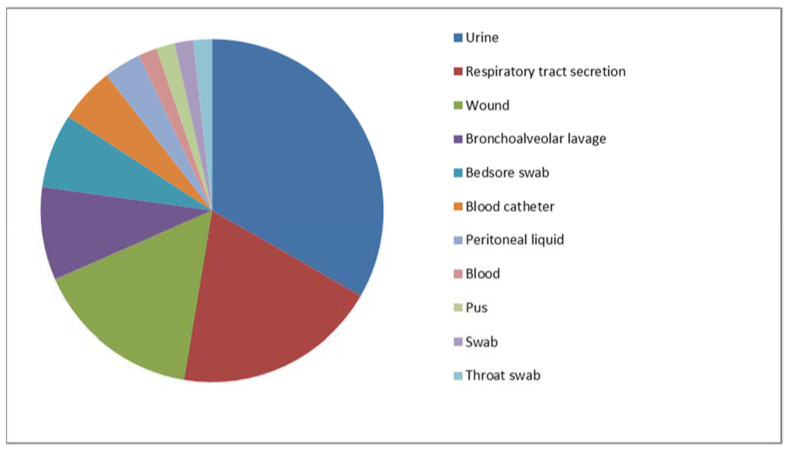
The origins (specimen types) of the multidrug-resistant *P. aeruginosa* strains included in the study (*n* = 57).

**Figure 5 ijms-24-01269-f005:**
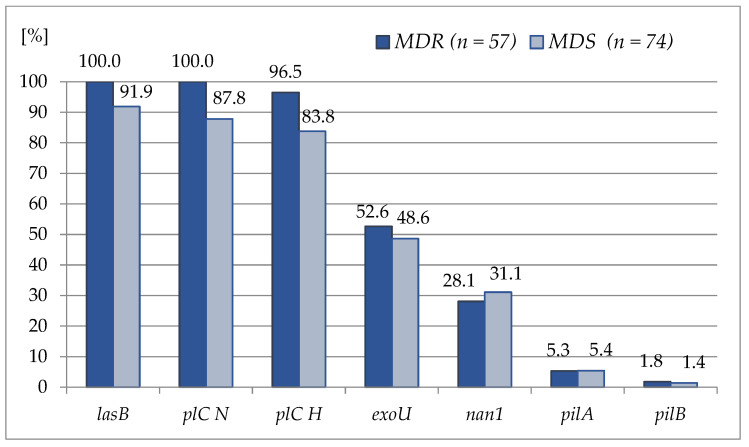
Prevalence of the studied virulence genes amongst the *P. aeruginosa* strains included in the study with respect to the strains’ antimicrobial susceptibility profiled (multidrug-resistant, MDR, *n* = 57 vs. multidrug-sensitive, MDS, *n* = 74).

**Table 1 ijms-24-01269-t001:** Characteristics and distribution of the genotypes detected amongst the examined multidrug-sensitive *P. aeruginosa* strains (*n* = 74).

Genotype			Gene Presence (+) or Absence (-) in a Particular Genotype						
Name	*n*	*%*	*lasB*	*plC N*	*plC H*	*exoU*	*nan1*	*pilA*	*pilB*
I-S	1	1.4	+	+	+	+	+	-	+
II-S	10	13.5	+	+	+	+	+	-	-
III-S	15	20.3	+	+	+	+	-	-	-
IV-S	1	1.4	+	+	+	-	+	+	-
V-S	6	8.1	+	+	+	-	+	-	-
VI-S	2	2.7	+	+	+	-	-	+	-
VII-S	22	29.7	+	+	+	-	-	-	-
VIII-S	1	1.4	+	+	-	+	+	-	-
IX-S	2	2.7	+	+	-	+	-	-	-
X-S	2	2.7	+	-	+	+	+	-	-
XI-S	1	1.4	+	-	+	+	-	-	-
XII-S	1	1.4	+	-	+	-	+	+	-
XIII-S	1	1.4	+	-	+	-	+	-	-
XIV-S	2	2.7	+	-	-	+	-	-	-
XV-S	1	1.4	+	-	-	-	-	-	-
XVI-S	2	2.7	-	+	-	+	-	-	-
XVII-S	3	4.1	-	+	-	-	-	-	-
XVIII-S	1	1.4	-	-	-	-	-	-	-

(+)—presence of a particular gene, (-)—absence of a particular gene.

**Table 2 ijms-24-01269-t002:** Characteristics and distribution of the genotypes detected amongst the examined multidrug-resistant *P. aeruginosa* strains (*n* = 57).

Genotype			Gene Presence (+) or Absence (-) in a Particular Genotype						
name	*n*	*%*	*lasB*	*plC N*	*plC H*	*exoU*	*nan1*	*pilA*	*pilB*
I-R	1	1.8	+	+	+	+	+	+	-
II-R	8	14.0	+	+	+	+	+	-	-
III-R	20	35.1	+	+	+	+	-	-	-
IV-R	1	1.8	+	+	+	-	+	+	+
V-R	5	8.8	+	+	+	-	+	-	-
VI-R	1	1.8	+	+	+	-	-	+	-
VII-R	19	33.3	+	+	+	-	-	-	-
VIII-R	1	1.8	+	+	-	+	+	-	-
IX-R	1	1.8	+	+	-	-	-	-	-

(+)—presence of a particular gene, (-)—absence of a particular gene.

**Table 3 ijms-24-01269-t003:** Specification of the PCR primers and parameters applied in the present study.

Virulence Factor Detected	Gene/PCR Primer Name	Manufacturer	Primer Sequence 5′→3′	Tm (°C)	Annealing Temperature (°C)	Amplicon Size (bp)
Exotoxin U	*exoU F*	Sigma	CCGTTGTGGTGCCGTTGAAG	55.9	64	134
	*exoU R*		CCAGATGTTCACCGACTCGC	55.9		
Phospholipase C (non-hemolytic)	*plC N F*	Integrated DNA Technologies	GTTATCGCAACCAGCCCTAC	55.9	54	466
	*plC N R*		AGGTCGAACACCTGGAACAC	57.2		
Phospholipase C (hemolytic)	*plC H F*		GAAGCCATGGGCTACTTCAA	55.1	50	307
	*plC H R*		AGAGTGACGAGGAGCGGTAG	58.2		
Elastase B	*lasB F*	Genomed	GGAATGAACGAAGCGTTCTC	51.8	50	300
	*lasB R*		GGTCCAGTAGTAGCGGTTGG	55.9		
Pilin A	*pilA F*		ACAGCATCCAACTGAGCG	50.3	59	1675
	*pilA R*		TTGACTTCCTCCAGGCTG	50.3		
Pilin B	*pilB F*		TCGAACTGATGATCGTGG	48.0	56	408
	*pilB R*		CTTTCGGAGTGAACATCG	48.0		
Neuraminidase 1	*nan1 F*		AGGATGAATACTTATTTTGAT	42.6	47	1316
	*nan1 R*		TCACTAAATCCATCTCTGACCCGATA	56.4		

## Data Availability

The data presented in this study are available on request from the corresponding author.

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
