# Peer review of "Comparison of Virulence-Factor-Encoding Genes and Genotype Distribution amongst Clinical Pseudomonas aeruginosa Strains"

_ijms, 2023, doi:10.3390/ijms24021269_

Round 1
Reviewer 1 Report
Pseudomonas aeruginosa is associated with hospital-associated infections and pose serious risks to patients suffering from various diseases, such as cystic fibrosis and immune deficiencies. It is significant to compare the virulence factor-encoding genes between multidrug-sensitive and multidrug-resistant clinical strains. In this study, the authors selected 74 multidrug-sensitive and 57 multidrug-resistant clinical P. aeruginosa strains and 6 virulence factor-encoding genes lasB, plCH, plCN, exoU, nan1, pilA and pilB. It is found there is a higher prevalence of genes related to virulence factors in multidrug-resistance strains and lasB and plCN genes were present in a number of strains.
This topic is interesting. However, the analyses are preliminary. The virulence factor-encoding genes were not enough and more important virulence factors should be taken into consideration, such as rhaminolipid, pigment, polysaccharide and so on. In addition, the genotypes classification is confused, and it should be illustrated in detail in this text.
Author Response
REVIEWER #1
Reviewer: Pseudomonas aeruginosa is associated with hospital-associated infections and pose serious risks to patients suffering from various diseases, such as cystic fibrosis and immune deficiencies. It is significant to compare the virulence factor-encoding genes between multidrug-sensitive and multidrug-resistant clinical strains. In this study, the authors selected 74 multidrug-sensitive and 57 multidrug-resistant clinical P. aeruginosa strains and 6 virulence factor-encoding genes lasB, plCH, plCN, exoU, nan1, pilA and pilB. It is found there is a higher prevalence of genes related to virulence factors in multidrug-resistance strains and lasB and plCN genes were present in a number of strains.
This topic is interesting. However, the analyses are preliminary. The virulence factor-encoding genes were not enough and more important virulence factors should be taken into consideration, such as rhaminolipid, pigment, polysaccharide and so on. In addition, the genotypes classification is confused, and it should be illustrated in detail in this text.
Answer: The authors appreciate the commentaries on the work. The collection of isolates included in this work comprises 74 multidrug-sensitive and 57 multidrug-resistant P. aeruginosa clinical isolates and the presence in their genomes of 7 different genes related to virulence were analysed. Although we recognize that some phenotypic traits (e.g. pigments synthesis) might be analysed, typing of strains based on phenotypes is always dubious as the expression of phenotype is highly dependent on environmental factors. We therefore disagree on the inclusion of phenotypic traits, although we thank for the suggestion of including additional virulence factor-encoding genes in future work. We have made some modifications on the genotypes description. Moreover, the detected genes were chosen to cover a number of virulence factors specificity/virulence factors with different activity: typical toxins, most important enzymes as well as structure-dependent factors, including these well-known, as well as these investigated rather rarely.
Reviewer 2 Report
An interesting read.
Title: Authors might need to look into title modification. "multidrug sensitive and multidrug resistant". there might be need to modify.
Abstract: can be improved on to include some specific data on results.
Introduction:
Line 58-59: Sentence is hanging and not connecting
Line 65: repetition
Line 65-68: Basic microbiological knowledge. Is this necessary?
Materials and methods:
Line 295: All description of sample collection and duration should be relocated to the beginning of the section.
Line 301-303: any reference?
Line 305-307: kindly relocate
Line 318-321: sentence is too long. Modify for easy read
Line 337-342: Sentences are disjointed
Results:
Line 123: might need to relocate and placed were the results are described
Discussion:
throughout this section, references have not followed the Journal guidelines. Some have been highlighted on the manuscript. Authors need to check this out and maintained journal guidelines.
Generally discussion is too long with more analysis on findings by other researchers. There will be need for the authors to discuss more on their results while correlating them with those of others.
Author highlight the novelties in their research findings
Line 219-226: kindly relate to the present investigation
Line 224: "the study," do the authors mean their investigation?
Line 228-232: This is basic knowledge. There is need to correlate this with the research findings.
Kindly see additional comments on the reviewed manuscript.

Author Response
REVIEWER #2
Reviewer: An interesting read.
Title: Authors might need to look into title modification. "multidrug sensitive and multidrug resistant". there might be need to modify.
Answer: Thank you for this comment. We have modified the title as suggested. It now reads as follows: “Comparison of virulence factor-encoding genes and genotypes distribution amongst clinical Pseudomonas aeruginosa strains“
Reviewer: Abstract: can be improved on to include some specific data on results.
Answer: Thanks for the suggestion. We have made modifications on the abstract and relocated a sentence. It now reads as follows: “Pseudomonas aeruginosa is an opportunistic pathogen encoding in its genome several multiple virulence factors, well known for its ability to cause severe and life-threatening infections, in particular among cystic fibrosis patients. The organism is also major cause of nosocomial infections, mainly affecting patients with immune deficiencies, burn wounds, ventilator-assisted patients, and patients suffering from other malignancies. The extensively reported emergence of multidrug-resistant (MDR) P. aeruginosa strains pose additional challenges to management of infections. The aim of this study was to compare the incidence of selected virulence factor-encoding genes and the genotypes distribution amongst multidrug-sensitive (MDS) and MDR clinical P. aeruginosa strains. The study involved 74 MDS and 57 MDR Pseudomonas aeruginosa strains and the virulence factor-encoding genes lasB, plC H, plC N, exoU, nan1, pilA, and pilB. The genotypes distribution, with respect to the antimicrobial susceptibility profiles of the strains, was also analyzed. The lasB and plC N genes were present amongst a number of P. aeruginosa, including all MDR P. aeruginosa, suggesting that their presence might be used as a marker for diagnostic purpose. A high variety of genotypes distribution was observed among the investigated isolates, MDS and MDR strains exhibiting, respectively, 18 and 9 distinct profiles. A higher prevalence of genes determining virulence factors in MDR strains was observed in this study, but more research is needed on the prevalence and expression levels of these genes in additional MDR strains.”
Reviewer: Introduction:
Line 58-59: Sentence is hanging and not connecting
Answer: Thanks for the suggestion. We have deleted the sentence.
Reviewer: Line 65: repetition
Answer: Thanks for the observation. We have deleted the sentence and re-written the following sentence, which now reads as follows: “In P. aeruginosa, the genes encoding virulence factors may be located on the bacterial chromosome or on the additional genome - genetic elements like plasmids and pathogenicity islands.”
Reviewer: Line 65-68: Basic microbiological knowledge. Is this necessary?
Answer: Thanks for the observation. Although it sounds as basic microbiology knowledge, this might be useful for the general readers of the journal, and therefore we decided to keep it.
Reviewer: Materials and methods: Line 295: All description of sample collection and duration should be relocated to the beginning of the section.
Answer: Thank you for this remark, the mentioned part has been replaced.
Reviewer: Line 301-303: any reference?
Answer: The corresponding reference has been added.
Reviewer: Line 305-307: kindly relocate
Answer: Thank you for this remark, the mentioned part has been replaced.
Reviewer: Line 318-321: sentence is too long. Modify for easy read
Answer: Thanks for the suggestion. The sentence was modified and now reads as follows: “Briefly, Taq polymerase was used with a total activity of 1 U per reaction in a 1 x concentrated polymerase buffer with 1.5 mM MgCl2 (Go Taq G2 Polymerase, Promega, Germany, FirePol DNA Polymerase, Solis BioDyne, Estonia) and 200 μM deoxynucleotide triphosphates (Promega, Germany, Solis BioDyne, Estonia).”
Reviewer: Line 337-342: Sentences are disjointed
Answer: Thanks for the observation. The formatting was solved.
Reviewer: Results:
Line 123: might need to relocate and placed were the results are described
Answer: Thank you for this remark, the mentioned parts have been replaced.
Reviewer: Discussion: throughout this section, references have not followed the Journal guidelines. Some have been highlighted on the manuscript. Authors need to check this out and maintained journal guidelines.
Answer: We thank you for this observation. The references were modified to keep with the journal guidelines.
Reviewer: Generally discussion is too long with more analysis on findings by other researchers. There will be need for the authors to discuss more on their results while correlating them with those of others.
Answer: We appreciate this remark. We wanted to discuss all our results in the light of the studies of other researchers. Therefore, the Discussion is quite long, indeed, but we believe that in the current version of the manuscript we managed to delete unnecessary parts, discuss all of our results and correlate them with other researches.
Reviewer: Author highlight the novelties in their research findings
Answer: It has been included into firs paragraph of the Discussion section.
Reviewer: Line 219-226: kindly relate to the present investigation
Answer: The whole paragraph has been rewritten.
Reviewer: Line 224: "the study," do the authors mean their investigation?
Answer: Thanks for the observation. We have rephrased the sentence which now reads as “The study by Mitov et al. also…”.
Reviewer: Line 228-232: This is basic knowledge. There is need to correlate this with the research findings.
Answer: Thank you for this remark, the mentioned part has been deleted.
Reviewer: Line 299: is there a reason for this?
Answer: Thank you for this remark, we only wanted to underline that the antimicrobial susceptibility testing interpretation criteria might have changed during the study and we wanted to adjust and unify them but it resulted in a huge diversity of susceptibility patterns which discouraged us to perform any statistical analysis as a consequence. However, if you find this sentence irrelevant, we decided to delete it.
Reviewer: Kindly see additional comments on the reviewed manuscript.
Answer: We sincerely appreciate the annotated file. The comments were addressed in the revised version.
Reviewer 3 Report
The authors have conducted a very nice study and the representation of results is good. Before proceeding further, the authors should consider following points:
1. Title: No need to mentioned “multidrug sensitive or multidrug resistant” It can be more better “Comparison of virulence factor-encoding genes and genotypes 2 distribution amongst clinical Pseudomonas aeruginosa strains”
2. Abstract: Authors are advised to add 2-3 sentences about the background of P. aeruginosa. Probably about AMR and nosocomial infections.
3. Abstract Line 25: write full form of P. aeruginosa.
4. Line 30: replace multidrug resistant with MDR.
5. Line 29-32: This sentence should be moved at the end of abstract as conclusive remarks.
6. Keywords: Authors are advised to write keyword which are not present in title e.g., antimicrobial resistance; AMR; P. aeruginosa; phenotypic profiles; genotypic profiles etc.
7. Line 42-44: Rewrite the sentence.
8. Line 48: write full form of COVID-19.
9. Line 285-286: This should be moved to results and the origin of isolates should be provided in detail in the results.
10. In the result section, author should write about the antibiotic susceptibility patterns.
11. In table 2, the column Genotype/gene and below rows is not clear. Please provide the legends. The authors can add one more row above with two columns to make it clarify. The 1se row should be labelled as “isolates” and second as “Genotype/gene”.
12. Rather then to add a separate subsection of statistical analysis, it is better to add one more row in the table and add p-values there. The remaining description can be provided below the table.
13. Line 139-140: The first sentence is not clear and need to rewrite with a suitable reference.
14. In the methods sections, author should clarify that wither they took the isolated bacteria from their bacterial strain store bank or they isolated directly from the clinical samples of patients.
15. The authors are advised to write more about isolation and identification of bacterial isolates.
16. Line 305: How the authors collected strains from distinctive patients? I think it should be: “The strains were isolated from the various clinical samples”.
17. Line 305-307: This sentence is not clear, and I suggest to not mention here if there is no data.
Figures S1 to S7 needs to elaborate according the labelled wells.
Author Response
REVIEWER #3
Reviewer: The authors have conducted a very nice study and the representation of results is good.
Answer: The authors kindly appreciate your time and effort in reviewing the manuscript and this the general positive appreciation of the work.
Reviewer: Before proceeding further, the authors should consider following points: Title: No need to mentioned “multidrug sensitive or multidrug resistant” It can be more better “Comparison of virulence factor-encoding genes and genotypes distribution amongst clinical Pseudomonas aeruginosa strains”
Answer: Thank you for the suggestion. We have changed the title as suggested.
Reviewer: Abstract: Authors are advised to add 2-3 sentences about the background of P. aeruginosa. Probably about AMR and nosocomial infections.
Answer: We kindly appreciate your suggestions. Therefore, we added the following sentences to the Abstract, new lines 22 to 27:
Pseudomonas aeruginosa is an opportunistic pathogen encoding in its genome several multiple virulence factors, well known for its ability to cause severe and life-threatening infections, in particular among cystic fibrosis patients. The organism is also major cause of nosocomial infections, mainly affecting patients with immune deficiencies, burn wounds, ventilator-assisted patients, and patients suffering from other malignancies. The extensively reported emergence of multiple drug resistant (MDR) P. aeruginosa strains pose additional challenges to management of infections.
Reviewer: Abstract Line 25: write full form of P. aeruginosa.
Answer: Thank you for the suggestion. The modification was done.
Reviewer: Line 30: replace multidrug resistant with MDR.
Answer: Thanks for the suggestion. The modification was done.
Reviewer: Line 29-32: This sentence should be moved at the end of abstract as conclusive remarks.
Answer: We kindly appreciate the suggestion. The sentence was moved to the end of the Abstract.
Reviewer: Keywords: Authors are advised to write keyword which are not present in title e.g., antimicrobial resistance; AMR; P. aeruginosa; phenotypic profiles; genotypic profiles etc.
Answer: Thanks for the suggestions. We have modified the keywords. The new Keywords are: Multiple antibiotics resistance; Pseudomonas aeruginosa virulence genes; virulence factors encoding genes, virulence genes genotyping, Pseudomonas aeruginosa clinical isolates, Antibiotic resistance profile
Reviewer: Line 42-44: Rewrite the sentence.
Answer: Thanks for the suggestion. The sentence now reads as follows: “Due to its high genetic plasticity and metabolic versatility, P. aeruginosa is able to colonize and cause infections to a wide range of hosts, from unicellular organisms to humans.”
Reviewer: Line 48: write full form of COVID-19.
Answer: Thanks for the suggestion. We have substituted the sentence “e.g., during COVID-19” by “e.g., during the Coronavirus Disease-2019 pandemics”
Reviewer: Line 285-286: This should be moved to results and the origin of isolates should be provided in detail in the results.
Answer: Thanks for the suggestion. The sentence was moved to the Results section and deleted from Supplementary material.
Reviewer: In the result section, author should write about the antibiotic susceptibility patterns.
Answer: This is a very valuable point; initially we even wanted to discuss our results in terms of the exact antimicrobial susceptibility patterns of the isolates included in the study. Within the group, however, the patterns of susceptibility varied so much that we chose to omit this aspect of the study because it could not even be statistically investigated.
Reviewer: In table 2, the column Genotype/gene and below rows is not clear. Please provide the legends. The authors can add one more row above with two columns to make it clarify. The 1se row should be labelled as “isolates” and second as “Genotype/gene”.
Answer: The tables were reshaped/redone. We hope that they are clear now.
Reviewer: Rather then to add a separate subsection of statistical analysis, it is better to add one more row in the table and add p-values there. The remaining description can be provided below the table.
Answer: This is a very valuable remark, initially we wanted to present the mentioned data as you suggested. However, our statistical analysis was calculated for gene pairs in each case. That is why it is not possible to add it into the tables in a simple manner, e.g. using single raw only.
Reviewer: Line 139-140: The first sentence is not clear and need to rewrite with a suitable reference.
Answer: It has been reworded into: “P. aeruginosa strains for many years have been a frequent object of many studies, mostly due to their significant relevance in hospital acquired infections.”
Reviewer: In the methods sections, author should clarify that wither they took the isolated bacteria from their bacterial strain store bank or they isolated directly from the clinical samples of patients.
Answer: It has been specified that “After the identification step, the strains were stored in brain heart infusion with 20% glycerol at -80°C.”
Reviewer: The authors are advised to write more about isolation and identification of bacterial isolates.
Answer: It has been completed.
Reviewer: Line 305: How the authors collected strains from distinctive patients? I think it should be: “The strains were isolated from the various clinical samples”.
Answer: We want to apologize for this unfortunate phrase. Indeed, the strains were isolated from various clinical samples derived from different patients. Therefore, the final paragraph is as follows: “The strains were isolated from various clinical samples derived from different patients and additionally checked for their similarity using pulsed-field gel electrophoresis (data not shown). Our goal was to exclude undistinguishable restriction patterns and avoid investigating repeating isolates.”
Reviewer: Line 305-307: This sentence is not clear, and I suggest to not mention here if there is no data.
Answer: With this sentence we only wanted to mention that the possibility of testing repeating isolated was excluded by application of PFGE methodology, which in our opinion is of high relevance for the reliability of the results. However, according to your (and another Reviewer) suggestions, the mentioned sentenced has been reshaped and replaced into strains characteristic section.
Reviewer: Figures S1 to S7 needs to elaborate according the labelled wells.
Answer: All the figures have been corrected according to your suggestions.